# Negative Regulation of Autophagy during Macrophage Infection by *Mycobacterium bovis* BCG via Protein Kinase C Activation

**DOI:** 10.3390/ijms25063145

**Published:** 2024-03-09

**Authors:** Rafael Maldonado-Bravo, Tomás Villaseñor, Martha Pedraza-Escalona, Leonor Pérez-Martínez, Rogelio Hernández-Pando, Gustavo Pedraza-Alva

**Affiliations:** 1Centro de Investigación en Dinámica Celular, Instituto de Investigación en Ciencias Básicas y Aplicadas, Universidad Autónoma del Estado de Morelos (UAEM), Cuernavaca 62210, Morelos, Mexico; rafael.maldonado@uaem.edu.mx; 2Sección de Patología Experimental, Instituto Nacional de Ciencias Médicas y Nutrición Salvador Zubirán, Alcaldía Tlalpan 14080, Ciudad de México, Mexico; 3Laboratorio de Neuroinmunobiología, Departamento de Medicina Molecular y Bioprocesos, Instituto de Biotecnología, Universidad Nacional Autónoma de México (UNAM), Cuernavaca 62210, Morelos, Mexico; tomas.villasenor@ibt.unam.mx (T.V.); leonor.perez@ibt.unam.mx (L.P.-M.); 4CONAHCyT-Unidad de Desarrollo e Investigación en Bioterapéuticos (UDIBI), Escuela Nacional de Ciencias Biológicas, Instituto Politécnico Nacional, Alcaldía Miguel Hidalgo 11340, Ciudad de México, Mexico; martha.pedraza@udibi.com.mx

**Keywords:** *M. tuberculosis*, *M. bovis*, macrophage, PKC, autophagy

## Abstract

*Mycobacterium tuberculosis* (*Mtb*) employs various strategies to manipulate the host’s cellular machinery, overriding critical molecular mechanisms such as phagosome-lysosome fusion, which are crucial for its destruction. The Protein Kinase C (PKC) signaling pathways play a key role in regulating phagocytosis. Recent research in Interferon-activated macrophages has unveiled that PKC phosphorylates Coronin-1, leading to a shift from phagocytosis to micropinocytosis, ultimately resulting in *Mtb* destruction. Therefore, this study aims to identify additional PKC targets that may facilitate *Mycobacterium bovis* (*M. bovis*) infection in macrophages. Protein extracts were obtained from THP-1 cells, both unstimulated and mycobacterial-stimulated, in the presence or absence of a general PKC inhibitor. We conducted an enrichment of phosphorylated peptides, followed by their identification through mass spectrometry (LC-MS/MS). Our analysis revealed 736 phosphorylated proteins, among which 153 exhibited alterations in their phosphorylation profiles in response to infection in a PKC-dependent manner. Among these 153 proteins, 55 are involved in various cellular processes, including endocytosis, vesicular traffic, autophagy, and programmed cell death. Importantly, our findings suggest that PKC may negatively regulate autophagy by phosphorylating proteins within the mTORC1 pathway (mTOR2/PKC/Raf-1/Tsc2/Raptor/Sequestosome-1) in response to *M. bovis BCG* infection, thereby promoting macrophage infection.

## 1. Introduction

*Mycobacterium tuberculosis* (*Mtb*) is the primary cause of Tuberculosis (TB), a disease predominantly affecting the respiratory tract. Historically, TB has been the leading cause of death worldwide due to an infectious agent, with 1.5 million deaths reported globally in 2020. However, the emergence of the COVID-19 pandemic, caused by the SARS-CoV-2 coronavirus [1], temporarily shifted TB to the second position among global health concerns. Nonetheless, TB continues to pose a significant public health challenge, affecting populations worldwide.

TB is primarily transmitted through the air via inhalation of aerosols contaminated with *Mtb.* Within the alveolar region of the lungs, *Mtb* interacts with various receptors expressed on the plasma membranes of phagocytic cells, initiating its internalization [2]. This internalization process relies on diverse receptors, including CD43 [3,4], the Dectin-1 receptor [5], the mannose receptor, scavenger receptor, CD14, pulmonary surfactant protein A, and complement receptors, among others (reviewed in [6]). Notably, studies using mice lacking some of these receptors have demonstrated that they are not indispensable for mycobacterial invasion of macrophages [7]. While there is no known specific receptor solely responsible for *Mtb* internalization, it is well-established that most of these receptors interacting with *Mtb* activate the PKC signaling pathway (reviewed in [8]).

The PKC family comprises phospholipid-dependent serine/threonine kinases that play a crucial role in regulating numerous signaling pathways governing processes such as growth, proliferation, cell differentiation, macrophage activation, and phagocytosis [9]. *Mtb*, being an intracellular pathogen, has evolved strategies to manipulate host signaling pathways and cellular machinery to circumvent essential molecular mechanisms involved in its destruction, most notably, the fusion of phagosomes with lysosomes [10,11]. One mechanism inhibiting phagosome-lysosome fusion involves the coating of the phagosome with Coronin-1, triggering calcium flux induction. This, in turn, activates the calcium-dependent phosphatase Calcineurin, leading to phagosome-lysosome fusion inhibition [12]. Consequently, this mechanism may enable mycobacteria to escape from the phagosome into the macrophage’s cytosol, primarily by disrupting the phagosome membrane [11,13].

However, experimental evidence has demonstrated that macrophages may induce autophagy activation as an alternative defense mechanism in response to *Mtb* infection, aiming to restore mycobacterial clearance [14]. Autophagy, an evolutionarily conserved mechanism in eukaryotes, primarily maintains organismal homeostasis in response to environmental and cellular stress conditions [15]. Paradoxically, several reports have indicated that *Mtb* can inhibit autophagy through various mechanisms, allowing it to survive within macrophages [16,17,18,19]. Coronin-1 is also implicated in impairing autophagosome formation in alveolar macrophages [18]. Additionally, pharmacological inactivation of PKCα by the specific inhibitor Tetrandine has been shown to activate autophagy through a mechanism involving mTORC1 inactivation [20].

Conversely, recent research has revealed that PKC, during IFN-γ-mediated macrophage activation, phosphorylates Coronin-1 at serine (Ser) residues in the positions 9, 311, 356, and 412, essential for switching from phagocytosis to macropinocytosis, resulting in mycobacterial destruction [21]. This experimental evidence suggests that the PKC signaling pathway, contingent upon the macrophage’s activation state, may either promote or hinder mycobacterial survival and macrophage infection by modifying the phosphorylation pattern of PKC target proteins.

In this study, we assess the role of PKC signaling pathway targets in blocking the macrophage’s molecular mechanisms deployed to prevent mycobacterial invasion and survival in naive human THP-1 macrophages. Following a phosphoproteomic profiling approach, we identify that *M. bovis BCG* macrophage infection via PKC induces the phosphorylation of at least 153 proteins. Notably, 55 out of the 153 PKC-phosphorylated proteins are associated with diverse processes, including immune response, endocytosis, autophagy, programmed cell death, and cellular trafficking. Ultimately, our findings suggest that *M. bovis BCG* could negatively regulate autophagy through the mTORC2(Sin-1)-PKC-Ras-Raf1-ERK1/2-TSC2-mTORC1 (Raptor)-Sequestosome-1 pathway in a PKC-dependent manner, thereby promoting mycobacterial infection.

## 2. Results and Discussion

### 2.1. M. bovis BCG Infection Induces the Phosphorylation of Different PKC Substrates

To evaluate whether the PKC signaling pathway may favor mycobacterial survival in macrophage infection through the modification of PKC-mediated protein phosphorylation, we assessed the phosphorylation of PKC substrates in total cell extracts of THP-1 human macrophages infected at an MOI of 1 (Multiplicity of Infection:1) in the presence or absence of a general PKC inhibitor, Gö-6983, using a specific antibody that recognizes substrates phosphorylated by PKC.

Consistent with the fact that macrophages exhibit basal levels of active PKC [22,23], non-infected macrophages displayed several proteins with basal phosphorylation levels (Figure 1A, lane 3). However, when macrophages were exposed to *M. bovis*, the number of PKC-phosphorylated proteins increased 30 min post-infection, along with the phosphorylation levels of proteins that were already basally phosphorylated (Figure 1A, lane 4). PKC-phosphorylated proteins ranged from 250 KDa to proteins smaller than 25 KDa (Figure 1A). As expected, treatment of macrophages with PMA/Ionomycin led to an increase in de novo PKC-phosphorylated proteins and in the phosphorylation levels of basally phosphorylated proteins (Figure 1A, lane 1).

To determine whether protein phosphorylation is PKC-dependent, THP-1 human macrophages were pre-treated with a general PKC inhibitor (Gö-6983) before *M. bovis* infection. PKC inhibition with Gö-6983 (Figure 1A, lanes 2 and 5) substantially decreased phosphorylation levels, indicating that protein phosphorylation in THP-1 human macrophages infected with *M. bovis* is a PKC-dependent mechanism. The highest levels of PKC-dependent phosphorylation in response to mycobacterial infection were observed at 30 min post-infection, compared to uninfected cells, decreasing after 1, 2, and 4 h post-infection (Figure 1A). Consequently, we identified PKC-phosphorylated proteins upon macrophage infection at the 30-min time point and conducted 12 independent experiments in triplicate (Figure 1B,C). These results support previously published data indicating PKC activation in response to *M. tuberculosis* macrophage infection [23]

### 2.2. Identification of PKC-Phosphorylated Proteins

To identify macrophage proteins phosphorylated by PKC during *Mycobacterium bovis BCG* infection, we collected total protein extracts (2 mg) from 12 independent experiments involving unexposed and mycobacterium-exposed macrophages, both pre-treated or not with a general PKC inhibitor. As a positive control, we used protein extracts from TPA/Ionomycin-treated macrophages. Subsequently, we conducted an enrichment of phosphorylated peptides, which were identified through mass spectrometry (LC-MS/MS) (Figure 1C).

The mass spectra of peptide fragments were compared to the *Homo sapiens* protein repertoire using the Uniprot non-redundant database (www.uniprot.org) and then visualized using Scaffold Viewer 3.6.4 software. We considered carbamidomethylation of Cysteine (C) as a constant modification, while oxidation of Methionine (M), and phosphorylation of Serine, Threonine, and Tyrosine (S, T, and Y) residues as variable modifications. The maximum precision error was set at 0.6 Da per fragment ions.

We established the following criteria for protein identification: (1) The presence of at least two unique peptides for each protein; and (2) a False Discovery Rate (FDR) of 0.1%. In total, we identified 736 phosphorylated proteins across all samples (unstimulated; BCG; BCG/Gö-6983; and PMA/Ionomycin), with 635 of these proteins present in three of the four samples (unstimulated, BCG, and BCG/Gö-6983) (Figure 2A). These proteins are highlighted in yellow in the Venn diagram in Figure 2A.

### 2.3. Phosphoproteomic Analysis

To assess changes in protein phosphorylation during macrophage infection with *M. bovis BCG* and to define PKC-dependent phosphorylation events, we conducted a comparative analysis of the phosphoproteome.

Out of the 736 identified phosphorylated proteins using Scaffold Viewer software, we filtered them so that each protein had to be present in at least three of the four experimental conditions (unstimulated cells, mycobacterium-infected cells in the absence or presence of the PKC inhibitor), and each protein should have the same identified peptide to compare changes in phosphorylation between the same peptide (Figure 2B). After applying these parameters, we identified proteins for which phosphorylation increased upon macrophage infection, but this increase was reversed when macrophages were infected in the presence of the PKC inhibitor (Figure 2B,C).

The analysis of the phosphorylated peptides identified in mass spectrometry involved comparing the sequence of each protein across the four conditions. Figure 2B illustrates this analysis for Syntaxin 4 protein; out of two identified peptides (highlighted in yellow, as identified in mass spectrometry), one displayed change in phosphorylation (phosphorylated residues marked in green as S, T, or Y) (Figure 2B).

For example, the identified peptides of the Syntaxin 4 protein showed phosphorylation at Ser 14, Ser 15, and Ser 117 residues in unstimulated cells (Figure 2B, black box). Upon infection with mycobacterium, the peptide AIEPQKEEADNYNSVNTR retained basal phosphorylation at residue Ser 117 while phosphorylation on Thr 120 was induced (Figure 2B, red box). In contrast, when macrophages were infected in the presence of the general PKC inhibitor, Thr 120 phosphorylation was lost, indicating that Syntaxin 4 phosphorylation on Thr 120 was induced in response to mycobacteria in a PKC-dependent manner (Figure 2B, blue box). However, phosphorylation of Ser 14, Ser 15, and Ser 117 remained unchanged when PKC activity was inhibited, suggesting the involvement of a different kinase for basal phosphorylation of Syntaxin 4 on these residues.

Out of the 736 identified phosphorylated proteins, 153 were present in three out of the four samples and exhibited changes in peptide phosphorylation in response to infection that reversed when the general PKC inhibitor was used (Figure 2C). Therefore, our phosphoproteomic analysis identified 153 proteins that are phosphorylated in response to *M. bovis BCG* infection in a PKC-dependent manner. This conclusion is grounded in the observation that, at the concentration used, the PKC inhibitor Gö-6983 has not been reported to effectively inhibit other kinase families. Additionally, the phosphorylated amino acids identified were located within the consensus sequence for PKC phosphorylation.

Upon further analysis using online Reactome software version 83, we found that these proteins are associated with biological processes such as immune response, signal transduction, cell cycle, gene expression, programmed cell death, and cell trafficking (Figure 2D) [24]. These results provide an overview of key biological processes regulated by the PKC signaling pathway in macrophages during the establishment of mycobacterial infection.

Additionally, when we conducted a more detailed analysis using the Gene Ontology and Uniprot databases, we identified 55 proteins (Appendix A) out of the 153 that are involved in biological processes such as endocytosis, vesicular trafficking, autophagy, and programmed cell death. These processes are often hijacked by mycobacteria to establish a successful infection [2,25], correlating with the pivotal role of PKC in regulating these processes [9].

### 2.4. String Analysis

Utilizing the STRING-12.0 database, we conducted an analysis of the interaction between the 55 proteins to identify PKC targets that could potentially promote macrophage infection by *M. bovis BCG* (see Figure 3).

In this context, we identified two distinct groups of proteins with robust interactions. These interactions imply the activation of specific signaling pathways. One group suggests the induction of phagocytosis of mycobacteria (Figure 3), while the other is involved in the regulation of autophagy (Figure 3) (refer to Table 1). Further elaboration on the interactions among these proteins in the context of these processes will be provided later.

### 2.5. PKC Substrates Involved in the Signaling Pathway Regulating Endocytosis

Phagocytosis is a fundamental process that modulates the immune response, involving the uptake of foreign particles, pathogens, and dead cells with a size larger than 0.5 μm [26,27]. Upon recognition by membrane receptors, these particles are internalized into phagosomes for subsequent elimination through fusion with lysosomes, where microorganisms are typically neutralized via reactive oxygen and nitrogen species as well as proteolytic enzymes [6].

Among the proteins identified, a subset exhibited strong interactions with each other. Notably, the protein kinase Raf-1, known for its role as a regulator between Ras GTPases and the MAPK/ERK signaling pathway, displayed robust interactions with Lyn, SHIP-1, and Dynamin 1. Raf-1, acting as a switch for various cellular processes, links Ras GTPases to the MAPK/ERK pathway, influencing cell fate decisions such as proliferation, differentiation, apoptosis, survival, and oncogenic transformation [28]. The protein Lyn, a tyrosine-protein kinase, is pivotal in transmitting signals from cell surface receptors to intracellular targets, playing a significant role in the regulation of both innate and adaptive immune responses. Additionally, Dynamin 1 and Dynamin 2, microtubule-associated proteins involved in vesicle separation during receptor-mediated endocytosis, displayed strong interactions. These interactions extended to the SH3 domain of the adaptor protein SH3KBP1 (SH3 kinase-binding protein). SH3KBP1 is associated with multiple signal transduction pathways, particularly participating in endocytosis and lysosomal degradation of ligand-induced receptor tyrosine kinases, and plays a vital role in cell morphology and cytoskeleton organization. Lyn, Raf-1, SH3KBP1, and Dynamin 1 also exhibited interactions with the phosphatidylinositol phosphatase SHIP-1, which primarily hydrolyzes PIP3 phosphate-5 to produce PIP2, thereby negatively regulating the PI3K (phosphoinositide 3-kinase) pathways [29,30]. Notably, PIP2 is crucial in phagosome formation and maturation.

Another hub of strong interactions involved Gbf1, ARF-GEF2, and Smap2 proteins (Figure 3). Gbf1, a Guanine-nucleotide exchange factor (GEF), is responsible for initiating the coating of nascent vesicles in the early secretory pathway [31]. Smap2, a GTPase acting on ARF1 and ARF6, contributes to the integrity of the endosomal compartment and regulates Golgi vesicular transport, particularly involved in trafficking from the trans-Golgi network to endosomes. The interactions between these proteins suggest the activation of a signaling pathway that potentially enhances phagocytosis of mycobacteria, thereby facilitating the invasion of macrophages by the pathogen.

In the context of proteins associated with the signaling pathway governing mycobacterial internalization, we observed that *M. bovis BCG* induced phosphorylation of Lyn at Ser 11, and this event was found to be PKC-dependent (see Appendix A). Although the phosphorylation of this site has been documented in the PhosphositePlus database, its physiological significance remains unknown. Considering Lyn’s position within the tyrosine kinase family, its regulation at Ser/Thr residues has received limited study. Nevertheless, we cannot rule out the possibility that the observed Lyn phosphorylation may contribute to the maintenance of PKC phosphorylation, as PKC is a known substrate of Lyn. This interplay might enable PKC to induce phosphorylation of Raf-1, as previous studies have indicated that PKC can directly phosphorylate Raf-1 at several sites (Kolch et al., 1993). Alternatively, PKC may induce Raf-1 phosphorylation indirectly.

Raf-1, a Ser/Thr kinase involved in regulating signal transduction and diverse cellular processes, exhibits several basal phosphorylation events, including Ser 621, Ser 43, Ser 259, and Ser 499. However, upon macrophage infection with mycobacteria, additional phosphorylation events at Thr 49 and Ser 257 were induced (see Appendix A). Importantly, these phosphorylation events were prevented by PKC inhibition. While phosphorylation at these sites has been reported in various cancer types (PhosphositePlus), their functional significance remains unclear. Specifically, Ser 257 resides within the CR2 region, a Ser/Thr-rich region containing binding sites for 14-3-3 proteins [32]. Phosphorylation at Ser 259 enables binding to 14-3-3 proteins, negatively regulating Raf-1 activity [33]. Conversely, it has been described that Ras binding to Raf-1 induces phosphorylation at Ser 257, an essential event for promoting Raf-1 activity [34]. This dynamic may elucidate the interaction observed in the interactome between Lyn kinase and Raf-1, where Lyn triggers Ras binding to Raf-1, leading to its phosphorylation and activation of the downstream MAPK Kinase pathway, ultimately resulting in cytoskeleton remodeling and phagosome formation.

As mentioned earlier, we identified interactions between Raf-1 and Dynamin 1, which in turn interacts with Dynamin 2. Dynamin proteins play a pivotal role in endocytosis, mediating the budding and cleavage of newly formed vesicles and endosomes on cell surfaces and within cell organelles, thereby influencing pathogen infection [35]. In our phosphoproteomic study, we observed that Dynamin 1 was phosphorylated at Thr 776 and Thr 780 in uninfected macrophages. However, upon infection, Dynamin 1 underwent dephosphorylation through a PKC-dependent mechanism, an effect reversed by PKC inhibition. Additionally, infection-induced phosphorylation of Dynamin 1 at Ser 777 was also PKC-dependent (see Appendix A). Research has indicated that Dynamin 1 is phosphorylated by PKC [36,37] and dephosphorylated by Calcineurin in terminal nerves, inhibiting dynamin GTPase activity [38]. Notably, dephosphorylation of Dynamin 1 by Calcineurin is essential for endocytosis in terminal nerves, suggesting a feedback loop in Dynamin 1 function regulated by PKC kinase and Calcineurin phosphatase.

Concerning Dynamin 2, we observed that upon infection phosphorylation occurred at Ser 761 and Ser 764, with these effects being reversed in the presence of a PKC inhibitor. Previous reports have indicated the involvement of Dynamin 2 in mycobacterial phagocytosis [39]. Additionally, studies have shown that Dynamin 2 is recruited to early phagosomes containing mycobacteria, facilitating the recruitment of Golgi-derived vesicle mannosidase 2 in Raw 264.7 and THP-1 macrophages [40]. Nevertheless, reports suggest that phosphorylation of Dynamin 2 at Ser 764 in bone marrow-derived macrophages inhibits its activity [38]. Nonetheless, changes in phosphorylation of this protein likely promote the release of the phagosome from the cell membrane, thereby enabling the internalization of mycobacteria into macrophages. This is supported by the interaction between Raf-1 protein and dynamin proteins (Figure 3), where Raf-1 may induce cytoskeleton reorganization for phagosome formation and trafficking.

Additionally, we observed interactions between dynamins and SH3BKP1, also known as CIN85, an adaptor protein that interacts with c-Cbl proteins exhibiting ubiquitin ligase activity. SH3BKP1 plays a role in negatively regulating signaling through ubiquitination, internalization, and degradation of activated receptors [41]. Interestingly, we found that infection with mycobacteria induced phosphorylation of SH3BKP1 at Ser 649, an effect reversed in the presence of a PKC inhibitor (see Appendix A). Although the physiological consequences of this phosphorylation have not been reported, previous studies in human neutrophils stimulated with PMA have shown that PKC induces SH3BKP1 phosphorylation, positively regulating the ubiquitination and degradation of the FcγRlla receptor [42]. Notably, we did not observe any effect on SH3BKP1 when cells were stimulated with PMA. In this context, it is possible that mycobacteria induce the internalization of the receptor to which they bind, ensuring their internalization into macrophages and establishing infection.

In summary, these results collectively suggest the activation of a signaling pathway that likely promotes phagocytosis of mycobacteria, thus facilitating macrophage invasion by these pathogens.

### 2.6. PKC Substrates Involved in the Signaling Pathway Regulating Autophagy: A Possible Alternative Pathway Induced by Mycobacteria to Promote Macrophage Infection

Autophagy is a highly conserved intracellular catabolic degradation process that plays a vital role in maintaining cellular homeostasis [43]. This process encompasses the formation of double-membrane vesicles known as autophagosomes, which sequester damaged cytosolic components, proteins, and organelles for subsequent degradation through fusion with lysosomes [43]. The breakdown products resulting from this degradation, including amino acids and fatty acids, are recycled, and employed for energy production and protein synthesis [41,44,45]. Additionally, autophagy can be activated by various endogenous and environmental stimuli, such as starvation, IFN-γ, rapamycin, hypoxia, oxidative stress, and radiation [46,47], as well as by the invasion of intracellular pathogens [14].

The autophagic process consists of several distinct steps, with key proteins and adaptors playing crucial roles. These steps include initiation, phagophore formation, phagophore elongation, lysosomal fusion or maturation, and degradation-recycling [48]. Specifically, autophagosome formation relies on a subset of proteins categorized into four groups:ULK Complex: Comprising ULK, ATG13, FIP200, and ATG101 proteins, this complex receives signals from mTOR and AMP-activated protein kinase (AMPK) and is instrumental in autophagosome initiation;Class III PI3K Complex: Consisting of Vps34-Beclin1 and Atg14 proteins, this complex marks membranes for nucleation and autophagosome generation;Ubiquitin-Like Protein Conjugation Systems: Comprising the Atg12 protein and the LC3 protein, these systems are involved in membrane formation and elongation;Transmembrane Proteins: ATG9 and VMP1, which are essential for membrane supply and mediate the binding of small membranes to forming autophagosomes, a critical step in their maturation [43,47,48].

Autophagy is finely regulated by the protein kinase AMPK, a cellular energy sensor [47], and mTORC1, a cell growth regulator that negatively controls autophagy [43]. The kinase activity of mTORC1 is strictly modulated by cellular nutrient levels. Nutrient presence stimulates mTORC1 activation, inhibiting autophagy. Conversely, nutrient deprivation leads to mTORC1 inactivation, thereby promoting autophagy.

Our phosphoproteomic analysis revealed that some of the proteins involved in autophagy displayed PKC-dependent changes in their phosphorylation patterns in response to mycobacterial infection. Notably, we observed strong interactions between certain proteins involved in autophagy regulation. Specifically, we identified an interaction between the MAPKAP1 protein, also known as Sin1, and the Tsc2 protein (Figure 3). Sin1 is a subunit of mTORC2 involved in regulating cell growth and survival in response to hormones, while Tsc2 forms a complex with Tsc1 and acts as a negative regulator of mTORC1, a primary regulator of cell growth and autophagy. Additionally, both proteins interacted with the Raptor protein (Figure 3), which controls mTORC1 activity. These proteins further interacted with Sequestosome-1 (Figure 3), an adaptor protein also known as ubiquitin-binding protein p62, which is essential for autophagosome formation and the autophagic degradation of polyubiquitinated bodies [49,50].

Given the involvement of these proteins in autophagy regulation and their PKC-dependent phosphorylation changes in response to mycobacterial infection, it is worthwhile to investigate the detailed interaction between PKC and these proteins. Autophagy serves as a defense mechanism in macrophages to facilitate the degradation of mycobacteria [14] but could also potentially represent an alternative pathway exploited by mycobacteria to enhance macrophage infection. Several reports suggest that *Mycobacterium tuberculosis* (*Mtb*) can inhibit autophagy through different mechanisms to ensure its survival within macrophages [16,17,18,19].

Previously, it was reported that pharmacological inactivation of PKCα using the specific inhibitor Tetrandine leads to autophagy activation through a mechanism involving mTORC1 inactivation [20]. Thus, it is plausible that *M. bovis BCG* activates a signaling pathway involved in protein synthesis and autophagy regulation independent of the classical AKT pathway. We propose that *M. bovis BCG* may modulate the mTORC2(Sin1)-PKC-Raf1-TSC2-mTORC1(Raptor)/Sequestosome-1 signaling pathway (Figure 4). The roles of individual PKC targets will be discussed further.

#### 2.6.1. Sin1 as a Key Factor in PKC Pathway Activation

Sin1, also known as SAPK-interacting protein or MAPKAP1, is a protein kinase that plays crucial roles in signal transduction, regulating the actin cytoskeleton in various organisms, and participating in cellular stress responses [51]. Initially identified as a protein interacting with ERK2 and Ras [52], Sin1 contributes to the negative feedback regulation of Ras activity, thereby suppressing Ras signaling [51] and controlling the dimerization and activation of ERK2 [53].

Moreover, Sin1 is a key component of the mTORC2 complex, playing a pivotal role in downstream signaling regulation. This complex responds to growth factors, influencing cell survival, metabolism, and cytoskeleton reorganization [54]. mTORC2 positively regulates processes like phosphorylation and activation of AGC family kinases, including AKT and PKC [55,56]. For instance, mTORC2 induces the phosphorylation of PKCα at Ser 657, promoting cell growth, actin cytoskeleton reorganization, and downstream signaling (PhosphoSitePlus). Sin1’s interaction with PKC (zeta) through its pleckstrin homology domain (PH) also aids in its translocation to the membrane [56].

In our phosphoproteomic analysis, Sin1 phosphorylation was observed at Ser 510 and Ser 512 upon infection. Interestingly, these phosphorylations were found to be PKC-independent (Appendix A). Although the functional significance of these specific sites has not been reported, other phosphorylation sites on Sin1 have been linked to various cellular processes, such as apoptosis inhibition, cell growth induction, and regulation of signaling pathways (PhosphositePlus.org).

Notably, Sin-1 phosphorylation at Thr 509 in uninfected macrophages was reversed upon infection via a PKC-dependent mechanism (Appendix A). This effect could be mediated by phosphatidylinositol phosphatase SHIP-1, which negatively regulates PI3K signaling by hydrolyzing PIP3 to PIP2 [22]. Alternatively, the serine/threonine phosphatase PTEN, which primarily targets PIP3 (phosphate-3), necessary for PI3K and mTORC2 activation [22], could be involved. These phosphatases likely dephosphorylate Sin-1, affecting its kinase activity and down-regulating the PI3K/AKT signaling pathway. This aligns with experimental evidence showing Sin-1 contributes to negative feedback regulation of Ras activity [51]. Thus, Sin1 might play a crucial role in a PKC-mediated signaling activation loop triggered by mycobacterial infection. Altogether, our results suggest that the phosphorylation change at Thr 509 in response to mycobacteria could be a new site of PKC-mediated regulation of mTORC2 activity.

#### 2.6.2. Activation of Raf-1 in Response to *M. bovis* Infection

Raf-1, also known as MAP3K, is a protein kinase that acts as a regulator between Ras GTPases and the MAPK/ERK signaling pathway. This linkage plays a critical role in determining cell fate decisions, including proliferation, differentiation, survival, and oncogenic transformation [28].

In our study, Raf-1 phosphorylation at Thr 49 and Ser 257 was induced upon macrophage infection and reversed by a general PKC inhibitor (Appendix A). These specific residues are not within a consensus PKC phosphorylation sequence, suggesting that PKC indirectly regulates their phosphorylation. Previous research has indicated that Ras binding to Raf-1 is essential for inducing Ser 257 phosphorylation and promoting Raf-1 activity [34]. Therefore, our results suggest that Ser 257 phosphorylation in response to *M. bovis* infection might be a result of Ras binding to Raf-1, subsequently activating downstream ERK1/2 MAP kinases, as previously reported [23]. This connection could explain the interaction between Raf-1 and Tsc2 within the interactome, with Tsc2 potentially being inhibited by ERK1/2 MAP kinases following phosphorylation of Raf-1 at Ser 257. 

#### 2.6.3. PKC-Dependent Tsc2 Inhibition Leading to mTORC1 Activation

Tuberin (Tsc2), when interacting with hamartin (Tsc1), forms the TSC protein complex, which controls cell growth. This complex negatively regulates mTORC1, the main regulator of anabolic cell growth [57]. Tsc2 inactivation through phosphorylation leads to mTORC1 activation. While mTORC1 is mainly activated by the PI3K/AKT signaling pathway, other pathways like MAP kinase ERK can also regulate it [58] through phosphorylation of Raf protein [59].

We found that Tsc2 displayed basal phosphorylation at Ser 1365, which was reversed upon macrophage infection, and this effect was restored by a PKC inhibitor (Appendix A). Additionally, Ser 1375 phosphorylation was induced upon infection, again being reversed in the presence of a PKC inhibitor (Appendix A). Although the physiological functions of these specific sites have not been described, some studies suggest that Tsc2 phosphorylation inhibits autophagy. For example, phosphorylation at Ser 981 inhibits autophagy (PhosphositePlus). Moreover, Tsc2 can be phosphorylated by PKC delta at Ser 932 and Ser 939, leading to Tsc2 inactivation and subsequent mTORC1 activation in response to translation inhibitors [58]. PKC inhibition has been shown to suppress multisite phosphorylation of Tsc2 at Ser 664, Ser 939, and Thr 1462, preventing mTORC1 activation [60]. Hence, the induced phosphorylation of Tcs2 in THP-1 macrophages during mycobacterial infection could activate mTORC1 through a PKC-mediated mechanism. Altogether, our results suggest that phosphorylation at Ser 1375 in response to mycobacteria could be a novel site for regulating Tsc2 activity, leading to activation of mTORC1, and subsequently promoting autophagy inhibition.

#### 2.6.4. Phosphorylation of Raptor by PKC and Its Potential Effect on Autophagy Regulation

Raptor (Regulatory Associated protein of MTOR Complex 1) is a protein that regulates mTORC1 activity, a complex that controls cell growth, survival, and autophagy in response to nutrients and hormonal signals [54].

In our phosphoproteomic study, phosphorylation of Raptor at Ser 859 and Thr 865 was observed in response to infection, and these phosphorylations were reversed by a PKC inhibitor (Appendix A). Previous research has indicated that mTORC1-mediated phosphorylation of Raptor at Ser 863 promotes phosphorylation of Ser 859 and 855 [61]. These multisite phosphorylations, particularly Ser 863, induce Raptor’s interaction with mTORC1, leading to mTORC1 activation.

When mTORC1 is active, it iss recruited to the lysosomal surface [62,63], retaining transcription factors (TFE3 and TFEB) in the cytosol and blocking lysosomal biogenesis [64]. Additionally, Raptor’s interaction with the mTORC1 complex induces ULK-1 phosphorylation, inhibiting its activity and preventing autophagy progression [65]. Under starvation conditions, when mTORC1 is not activated, ULK-1 remains uninhibited, leading to autophagy activation [66]. Our data suggest that PKC-mediated phosphorylation at Ser 859 and Thr 865 of Raptor in response to mycobacteria could activate mTORC1, promoting the negative regulation of ULK-1 and inhibiting autophagy and lysosomal activity mediated by mTORC1 activation.

#### 2.6.5. Phosphorylation of Sequestosome-1/p62 by PKC and Its Potential Regulation of Autophagy

Sequestosome-1, also known as p62, is implicated in autophagosome formation and functions as a receptor for ubiquitinated proteins, organelles, and pathogens. These are sequestered and transported for degradation in the autophagosome [50,67].

In our phosphoproteomic analysis, p62 was found to be phosphorylated by PKC at Ser 275 in response to mycobacterial infection, and this effect was reversed in the presence of a PKC inhibitor (Appendix A). Intriguingly, this corresponds with preliminary experiments showing p62 accumulation and oligomerization in response to infection, which was also prevented by PKC inhibition (Appendix A). Accumulation of p62 is often associated with altered autophagic flux, and oxidative stress can lead to p62 oligomerization, affecting its ability to transport ubiquitinated cargo to the autophagosome [68]. Altogether, these results suggest that *M. bovis BCG* might be altering autophagic flux through p62 oligomerization, inhibiting autophagy and favoring macrophage survival and mycobacterial infection through PKC-mediated phosphorylation of p62 at Ser 275.

In conclusion, our data suggest that *M. bovis BCG* induces the activation of the mTORC1 pathway in a PKC-dependent manner, potentially contributing to the inhibition of autophagy. This proposed pathway of autophagy inhibition, mTORC2 (Sin-1)-PKC-Ras-Raf1-ERK1/2-TSC2-mTORC1 (Raptor)/Sequestosome-1, could be an essential mechanism for *M. bovis BCG* to evade clearance, thereby favoring infection and ensuring its survival within macrophages (Figure 4).

#### 2.6.6. Dephosphorylation of MAP-1B and Its Possible Regulation of Autophagosome Trafficking

MAP-1B, also known as Microtubule Associated Protein 1B, belongs to a protein family involved in microtubule assembly, crucial for neurogenesis [69]. It is essential for axon stability and acts as a positive cofactor in DAPK1-mediated autophagic vesicle formation [70]. Furthermore, MAP-1B can interact with LC3, facilitating phagosome-lysosome fusion. When MAP-1B is phosphorylated, it is associated with LC3-autophagosomes [71].

MAP-1B was found to be phosphorylated at Ser 828 and Ser 1387 in uninfected macrophages, but upon infection, MAP-1B was dephosphorylated by a PKC-dependent mechanism (Appendix A). While the physiological functions of these specific sites remain unclear, dephosphorylation of MAP-1B could potentially affect autophagosome trafficking due to its critical role in microtubule assembly.

### 2.7. Concluding Remarks

This study aimed to identify PKC targets that could enhance macrophage infection by *M. bovis BCG*. We conducted a phosphoproteomic analysis, revealing that the infection of THP-1 macrophages by *M. bovis BCG* activates PKC, leading to the phosphorylation of a minimum of 153 proteins. Among these, 55 proteins participate in various cellular processes, encompassing immune response, endocytosis, autophagy, apoptosis, and cell trafficking. Our findings suggest that *M. bovis BCG* initiates a signaling cascade culminating in the PKC-mediated activation of mTORC1, thereby imposing negative regulation on autophagy. These observations align with prior experimental studies demonstrating that *Mtb* inhibits autophagy through diverse mechanisms, effectively securing its survival within macrophages [16,17,18,19,72]. Considering that PKC is activated during macrophage infection by both *M. bovis* BCG and *M. tuberculosis,* it is possible that PKC in response to both pathogens promotes the phosphorylation of the same substrates to modulate autophagy. However, this remains to be tested experimentally. It is noteworthy that inhibition of PKC (specifically the alpha isoform) has been shown to promote autophagy activation [20]. 

Consequently, our data, reflecting alterations in protein phosphorylation patterns in response to *M. bovis* infection in macrophages, lead us to propose a novel signaling pathway of autophagy inhibition induced by *M. bovis BCG*: mTORC2 (Sin-1)-PKC-Ras-Raf-1-ERK1/2-TSC2-mTORC1 (Raptor)/Sequestosome-1. Ultimately, this mechanism could contribute to evading elimination via autophagy, thereby favoring infection, and ensuring the pathogen’s survival within the macrophage (Figure 4).

Finally, the data presented here open new avenues of research aiming at understanding the fine-tuned mechanisms through which a successful pathogen such as *Mycobacterium tuberculosis* ensures survival at the early stages of the infectious process. Defining specific signaling pathways activated by *M. tuberculosis* represents therapeutic targets to reactivate key biological processes involved in pathogen elimination. 

## 3. Material and Methods

### 3.1. Antibodies and Reagents

Primary antibodies, including anti-p(Ser)PKC substrates and anti-Sequestosome (#5114S), were purchased from Cell Signaling Technology (Danvers, MA, USA). Antibodies anti-pERK ½ (sc-7383) and anti-ERK 2 (sc-154) were obtained from Santa Cruz Biotechnology, Inc. (Santa Cruz, CA, USA). Horseradish peroxidase-coupled secondary antibodies anti-rabbit (Invitrogen, Waltham, MA, USA) and anti-mouse (Santa Cruz Biotechnology) monoclonal antibodies were used. The chemiluminescence kit employed for development was “Western Lightning Plus-ECL” from PerkinElmer Inc. (Waltham, MA, USA). Reagents used in cell culture, such as PMA (50 ng/mL and 20 ng/mL) and Ionomycin (1 μM), were purchased from Sigma-Aldrich (Burlington, Massachusetts, USA). The PKC inhibitor Gö-6083 was acquired from Calbiochem (Burlington, MA, USA). These reagents were dissolved in tissue culture-grade DMSO from Sigma-Aldrich.

### 3.2. Cell Line

Human THP-1 cells, obtained from the ATCC, were cultured in RPMI 1640 medium (Roswell Park Memorial Institute, GIBCO-Invitrogen (Waltham, MA, USA). The medium was supplemented with 10% heat-inactivated fetal bovine serum (FBS; Byproducts, Guadalajara, México), penicillin (100 μg/mL), streptomycin (100 μg/mL) (GIBCO-Invitrogen), and 2 mM glutamine. Cells were maintained at 37 °C with 5% CO_2_ and 90% humidity.

### 3.3. Cellular Differentiation and Activation

To induce differentiation of THP-1 cells into macrophages, 2.5 × 10^2^ cells were seeded in 35mm petri dishes and cultured at 37 °C with 5% CO_2_ and 90% humidity in the presence of PMA (50 ng/mL). After 24 h, undifferentiated, non-adherent cells were removed by washing with 1X PBS. The differentiated macrophages were then cultured for an additional 48 h, totaling 72 h of differentiation. The culture medium was changed to serum-free RPMI medium 12 h before activation. Differentiated THP-1 macrophages (2.5 × 10^5^) were exposed to *M. bovis BCG* (multiplicity of infection: MOI = 1), PMA (20 ng/mL), or Ionomicyn. When specified, cells were pre-incubated with the specific PKC inhibitor Gö-6983 (5 μM) for 30 min before activation.

### 3.4. Total Cell Extract Preparation, Electrophoresis, and Immunoblotting

Following activation, THP-1 cells were washed with PBS and lysed with lysis buffer (20 mM Tris [pH 7.4], 250 mM NaCl, 25 mM glycerophosphate, 25 mM pPiNa, 2 mM EDTA, 1% Triton X-100, 10% glycerol), supplemented with 1 mM phenylmethylsulfonyl fluoride (PMSF), 0.5 mM dithiothreitol (DTT), 200 mM Na3VO4, and a complete protease inhibitor cocktail (Roche, Basel, Switzerland). Cell lysates were then transferred to Eppendorf tubes and incubated for 10 min at 4 °C, followed by centrifugation (14,000 rpm, 10 min at 4 °C). The supernatant was stored at −70 °C until use. SDS-PAGE was used to separate proteins (20 μg), which were then transferred to nitrocellulose membranes and subjected to immunoblotting using a chemiluminescent substrate, ECL Plus (Perkin-Elmer), in accordance with the manufacturer’s instructions.

### 3.5. Trypsin Digestion

Phosphoproteomics experiments were conducted at the proteomics unit of the Institut de Recherches Cliniques de Montréal, Canada. For in-solution digestion, 1 mg of protein was reduced at 37 °C using dithiothreitol (DTT) for one hour and alkylated with iodoacetamide for 60 min at room temperature in the dark. The mixture was then digested using trypsin (enzyme/total protein ratio of 1:50) followed by incubation at 37 °C overnight. The tryptic digestion was quenched by adding 1% trifluoroacetic acid (TFA).

### 3.6. Desalting and Phosphopeptides Enrichment with TiO_2_

The resultant peptides were desalted using an Oasis HLB extraction plate (30 µm, Waters UK). The wells were equilibrated with 500 µL of 100% methanol and washed with 500 µL of H_2_O. Subsequently, wells were loaded with the peptide mixture, washed with 500 µL of 5% methanol, and eluted with 400 µL of 100% methanol. The recovered peptides were lyophilized and subjected to phosphopeptide enrichment as follows: Peptides were resuspended in 200 µL of 80% acetonitrile (ACN)/3% TFA/300 mg DHB (dihydroxybenzoic acid). TiO_2_ beads (GL sciences, Torrance, CA, USA) were resuspended in the same buffer, and 20 µL of this slurry was added to each sample (1:2 peptides to beads ratio). Samples were incubated for 30 min on a rotator at room temperature and then centrifuged at 5000× *g* for 1 min. Phosphopeptide-bound TiO_2_ beads were washed three times with 30% ACN, 3% TFA on a StageTip C8 material (ThermoFisher Scientific, Waltham, MA, USA), and then three times with 80% ACN, 0.3% TFA. Phosphopeptides were eluted twice using a C8 StageTip with 75 µL of 40% ACN, 15% NH4OH. Eluted phosphopeptide samples were vacuum-dried prior to LC-MS/MS analyses.

### 3.7. LC-MS/MS Analysis

Phosphopeptides enriched samples were re-solubilized under agitation for 15 min in 12 µL of 1% ACN/1% formic acid. The LC column used was a PicoFrit fused silica capillary column (17 cm × 75 µm i.d; New Objective, Woburn, MA, USA), self-packed with C-18 reverse-phase material (Jupiter 5 µm particles, 300 Å pore size; Phenomenex, Torrance, CA, USA) using a high-pressure packing cell. This column was installed on the Easy-nLC II system (Proxeon Biosystems, Odense, Denmark) and coupled to the LTQ Orbitrap Velos (ThermoFisher Scientific, Bremen, Germany), equipped with a Proxeon nanoelectrospray Flex ion source. The buffers used for chromatography were 0.2% formic acid (buffer A) and 100% acetonitrile/0.2% formic acid (buffer B). Peptides were loaded on-column at a flow rate of 600 nL/min and eluted with a two-slope gradient at a flow rate of 250 nL/min. Solvent B first increased from 1 to 37% over 100 min and then from 37 to 80% B over 20 min.

LC-MS/MS data acquisition was performed using a data-dependent top10 method combined with MS3 scanning upon detection of a neutral loss of phosphoric acid (48.99, 32.66, or 24.5 Th) in MS2 scans. The mass resolution for the full MS scan was set to 60,000 (at *m*/*z* 400), and lock masses were used to improve mass accuracy. The mass-to-charge ratio range was from 375 to 1800 for MS scanning with a target value of 1,000,000 charges, and from ~1/3 of the parent *m*/*z* ratio to 1800 for MS scanning in the linear ion trap analyzer with a target value of 10,000 charges. The data-dependent scan events used a maximum ion fill time of 100 ms, and target ions already selected for MS/MS were dynamically excluded for 30 s after 2 repeat counts. Nanospray and S-lens voltages were set to 1.5 kV and 50 V, respectively. The normalized collision energy used was 27 with an activation q of 0.25 and activation time of 10 ms. Capillary temperature was set at 250 °C.

### 3.8. Protein Identification

Peak list files were generated with Proteome Discoverer (version 2.1, Thermos Fisher Scientific, Waltham, MA, USA) using the following parameters: minimum mass set to 500 Da, maximum mass set to 6000 Da, no grouping of MS/MS spectra, precursor charge set to auto, and a minimum number of fragment ions set to 5. Protein database searches were conducted using Mascot 2.3 (Matrix Science, London, UK). The mass tolerance for precursor ions was set to 10 ppm, and for fragment ions, it was set to 0.6 Da. The specified enzyme was trypsin, and two missed cleavages were allowed. Cysteine carbamidomethylation was specified as a fixed modification, and methionine oxidation, serine, threonine, and tyrosine phosphorylation modifications were specified as variable modifications. Data analysis was performed using Scaffold (version 3.6).

## 4. Conclusions

Our data demonstrate that the PKC-mediated signaling pathways activated by *M. bovis* BCG during macrophage infection could play a crucial role in modulating essential biological processes such as endocytosis, cellular trafficking, programmed cell death, and autophagy. Given that PKC signaling is also modified during macrophage infection by *M. tuberculosis* strains, this underscores the significance of PKC as a potential therapeutic target to reactivate biological processes involved in pathogen elimination.

## Figures and Tables

**Figure 1 ijms-25-03145-f001:**
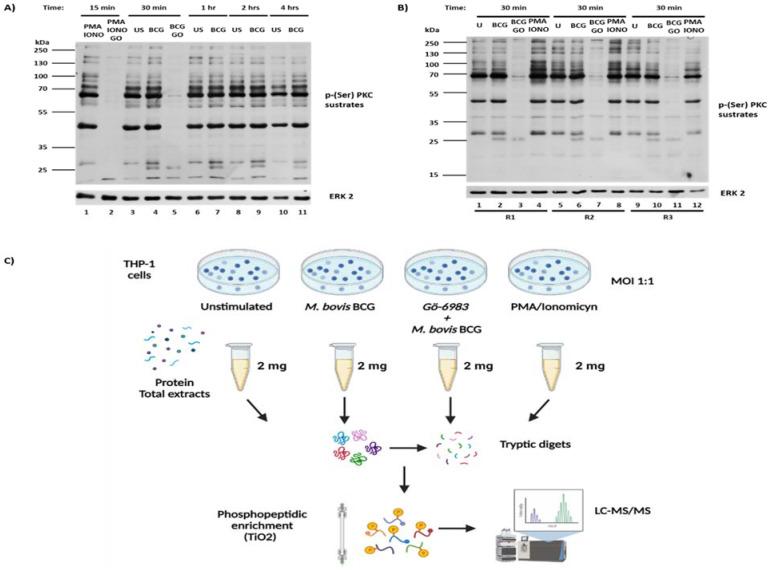
*M. bovis BCG* Infection Induces Phosphorylation of PKC Substrates Over Time. (**A**) Human macrophages (THP-1) were infected with *M. bovis BCG* (BCG) for varying durations: 30 min, 1 h, 2 h, or 4 h, in the presence or absence of the general PKC inhibitor (GO), Gö-6983 (5 µM). Unstimulated macrophages (US) served as a negative control, while cells treated with PMA (20 ng/mL) and Ionomicyn (1 μM) for 15 min served as a positive control (PMA-IONO). (**B**) THP-1 macrophages were infected with *M. bovis BCG* for 30 min in the presence or absence of the general PKC inhibitor, Gö-6983 (5 µM). Total lysates were separated via SDS-PAGE and probed with an anti-phosphorylated PKC substrates antibody. It is a representative image of twelve independent experiments in triplicate each; (R1) Replicate 1, (R2) Replicate 2, and (R3) Replicate 3. (**C**) Methodology employed for identifying the phosphoproteome of THP-1 macrophages in response to *Mycobacterium bovis BCG* infection. Panel created with BioRender.

**Figure 2 ijms-25-03145-f002:**
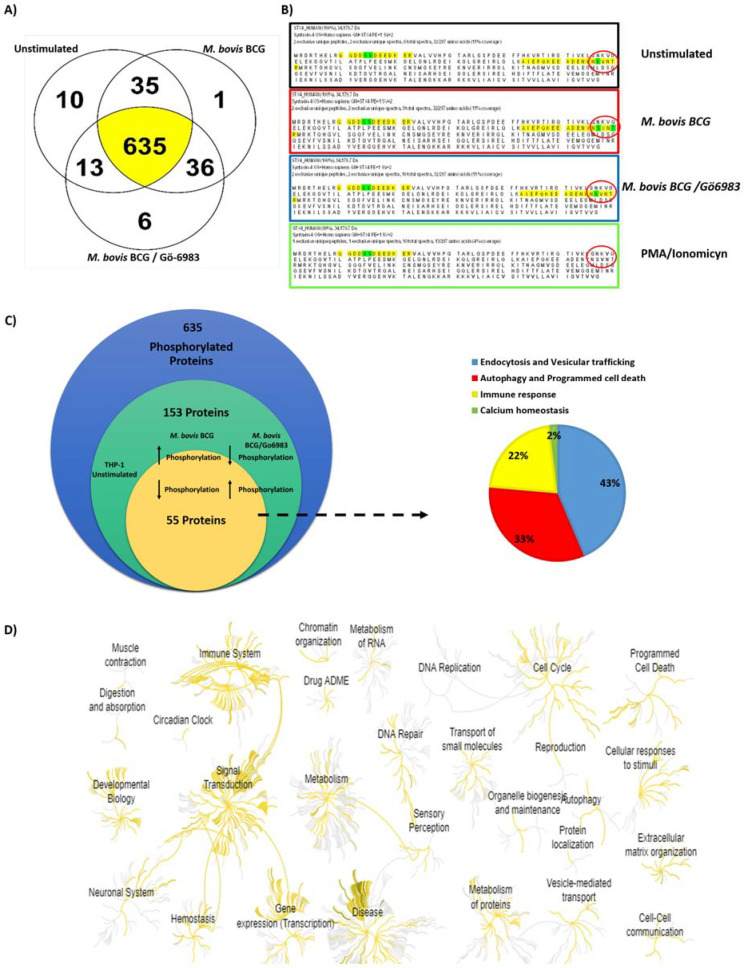
Phosphoproteomic Analysis. (**A**) The Venn diagram illustrates the number of proteins shared between different treatments, as provided by Scaffold Viewer software 4.8.8. (**B**) An example of comparative phosphoproteomic analysis is presented for Syntaxin 4. The protein sequence under four treatment conditions is depicted: black (stimulated cells), red (cells infected with *M. bovis BCG*), blue (cells treated with a PKC inhibitor), and green (cells treated with PMA/Ionomycin). Phosphorylated peptides identified in mass spectrometry are marked in yellow, and phosphorylation sites are highlighted in green (S, T). This example demonstrates the analysis process used for each protein within the group of 154 proteins. Syntaxin 4, involved in vesicular traffic, was selected as an illustration. (**C**) This schematic outlines the comparative analysis of the phosphoproteome in response to *Mycobacterium bovis* BCG infection in macrophages. The blue circle represents the 635 identified proteins, with parameters considered for phosphorylation changes. The green circle highlights the 153 proteins present in at least three of the four samples, displaying PKC-dependent changes in phosphorylation. The yellow circle represents proteins involved in processes such as endocytosis, vesicular traffic, immune response, autophagy, programmed cell death, and calcium homeostasis. This analysis was conducted manually. (**D**) Enriched biological processes of the 153 proteins displaying changes in phosphorylation in macrophages infected with *M. bovis BCG* are obtained from the Reactome platform. Yellow highlights the processes predominantly enriched in the phosphoproteome of macrophage infection, while non-enriched processes are shown in gray.

**Figure 3 ijms-25-03145-f003:**
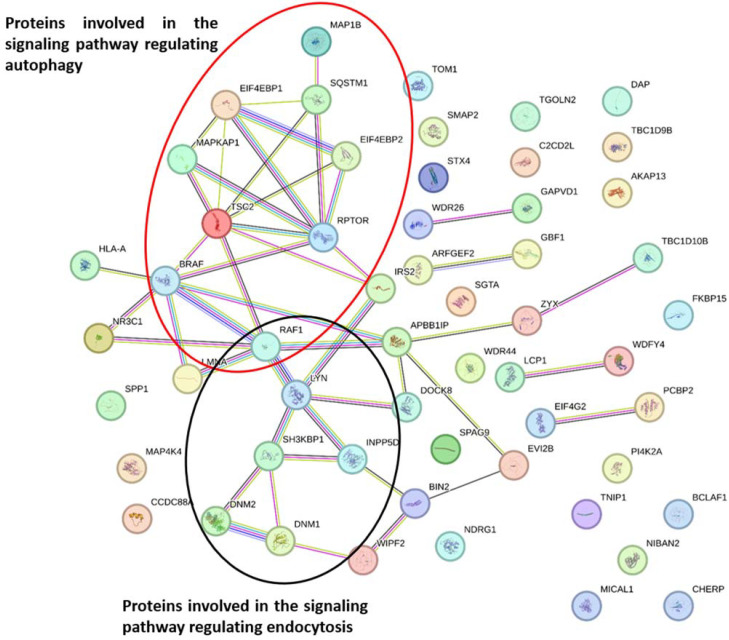
STRING Analysis of PKC-Phosphorylated Proteins in Response to *M. bovis BCG* Infection. The accession numbers of 55 proteins, which play pivotal roles in processes such as endocytosis, vesicular trafficking, immune response, autophagy, programmed cell death, and intracellular calcium homeostasis, were input into the STRING 12.0 platform. The STRING analysis unveiled a robust interaction network among these PKC-phosphorylated proteins, shedding light on the intricate molecular relationships governing their functions in response to *M. bovis BCG* infection.

**Figure 4 ijms-25-03145-f004:**
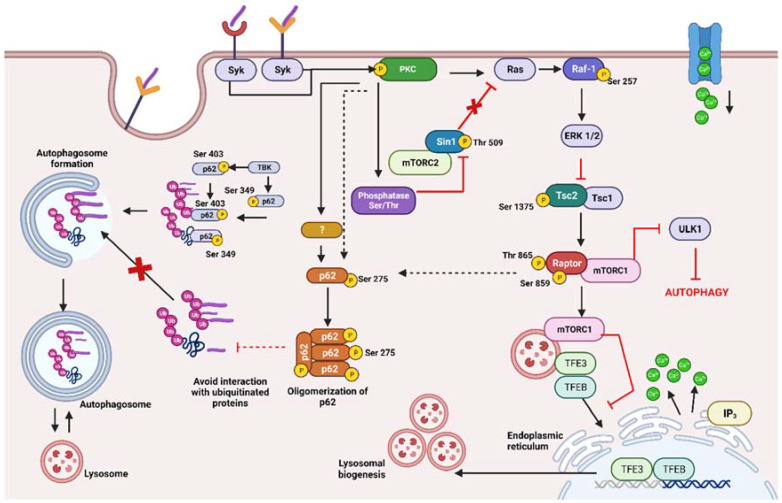
*Mycobacterium bovis BCG* negatively regulates autophagy through the activation of mTORC1 pathway (mTORC2(Sin-1)/PKC/Ras/Raf-1/ERK1/2/TSC2/mTORC1(Raptor)/Sequestosome-1), in a PKC-dependent manner, thus favoring mycobacterial infection. Created with BioRender.

**Table 1 ijms-25-03145-t001:** List of proteins involved in the autophagy negative regulation pathway and the site at which they are modified.

Protein	Gene	Accession	Phosphorylated Sites
Sin1	SIN1	Q9BPZ7	↓ Thr 509
Raf-1	RAF-1		↑ Thr49
		P04049	↑ Ser 257
Tsc2	TSC2	P49815	↓ Ser1365
			↑ Ser1375
Raptor	RPTOR	Q8N122	↑ Ser 859
			↑ Thr 865
Sequestosome-1	SQSTM1	Q13501	↑ Ser 275

Arrows indicate increase or decrease PKC-dependent phosphorylation upon *M. Bovis* macrophage infection.

## Data Availability

All data needed to evaluate the conclusions in the paper are present in the paper and/or the Appendix A. Additional data related to this paper may be requested from the corresponding author.

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
