# Peer review of "Negative Regulation of Autophagy during Macrophage Infection by Mycobacterium bovis BCG via Protein Kinase C Activation"

_ijms, 2024, doi:10.3390/ijms25063145_

Round 1

Reviewer 1 Report

Comments and Suggestions for Authors

The work is devoted to the comparison of phosphoproteomic profiles of BCG mycobacterium infected and non-infected cells of the THP-1 macrophage-like cell line.

The authors found differences in phosphoproteomic profiles, analyzed the functions of phosphorylation-dephosphorylation proteins and conclude "that PKC may negatively regulate autophagy by phosphorylating proteins within the mTORC1 pathway (mTOR2/PKC/Raf-1/Tsc2/Raptor/Sequestosome1) in response to M. bovis BCG infection, thereby promoting macrophage infection"

I have the following questions and comments

1.

 Judging from Fig1 the use of the Go 6983 inhibitor results in dephosphorylation of a significant portion of the protein. Is the phosphorylation of the protein mostly due to RBCs or is this an artifact of the experiment?

2. In all samples where there was BCG stimulation a 25 kDa band is present. Did you not do a MALDI analysis of this band?

3. There are errors in the caption of Fig1

Unstimulated macrophages (Ss) served as a negative control, while cells treated with TPA (20 ng/ml)

Methodology employed for identifying the phosphoproteome of THP-1 macrophages in response to Mycobacterium bovis BCG infection "Created with BioRender.com".

4. Figure 1A, lane 1.2.3.4.

" lane " not marked

5. Supplementary materials not provided

6. Caption to Fig3 "The accession numbers of 55 proteins". The names of proteins are given, not accession numbers. Very poor quality of the figure

7. The conclusion "Our findings provide compelling evidence that M. bovis BCG initiates a signaling cascade culminating in the PKC-mediated activation of mTORC1, thereby imposing negative regulation on autophagy" is not well supported. The authors demonstrate phosphorylation of certain proteins, followed by a good literature review of the functions of these proteins, and conclude compelling evidence based on this. The authors may suggest such a relationship, but do not claim that the evidence is compelling

Author Response

The work is devoted to the comparison of phosphoproteomic profiles of BCG mycobacterium infected and non-infected cells of the THP-1 macrophage-like cell line.

The authors found differences in phosphoproteomic profiles, analyzed the functions of phosphorylation-dephosphorylation proteins and conclude "that PKC may negatively regulate autophagy by phosphorylating proteins within the mTORC1 pathway (mTOR2/PKC/Raf-1/Tsc2/Raptor/Sequestosome1) in response to M. bovis BCG infection, thereby promoting macrophage infection"

I have the following questions and comments

  1. Judging from Fig1 the use of the Go 6983 inhibitor results in dephosphorylation of a significant portion of the protein. Is the phosphorylation of the protein mostly due to RBCs or is this an artifact of the experiment?

Thank you for your comment.

We consider these results to be genuine for several reasons: i) THP-1 macrophages are cultured in a complete medium containing various growth factors that activate PKC. Given that we are using an antibody binding to PKC-phosphorylated substrates to visualize PKC activity, it's not surprising to observe numerous proteins phosphorylated by PKC in unstimulated cells; ii) an increase in phosphorylation levels of certain proteins and the appearance of new proteins are evident upon PKC activation in PMA/Ionomycin-treated cells (positive control) and in response to the presence of BCG. The fact that the PKC inhibitor reduces the number of detected proteins indicates that most of the observed PKC substrates are phosphorylated by the α and β isoforms. The remaining phosphorylated proteins might be substrates for other PKC isoforms not affected by the inhibitor, or they could represent unspecific interactions of the antibody with phosphorylated amino acids resulting from the activity of other serine/threonine kinases.

  1. In all samples where there was BCG stimulation a 25 kDa band is present. Did you not do a MALDI analysis of this band?

    Thank you for your observation.

We also noticed the presence of this 25 kDa phosphorylated protein; however, given that this protein fallen in our exclusion criterion that is, the phosphorylation levels of this protein remained virtually unchanged in the presence of the PKC inhibitor, therefore no longer proceeded to a MALDI analysis.

  1. There are errors in the caption of Fig1

Unstimulated macrophages (Ss) served as a negative control, while cells treated with TPA (20 ng/ml)

Methodology employed for identifying the phosphoproteome of THP-1 macrophages in response to Mycobacterium bovis BCG infection "Created with BioRender.com".

Thank you very much for these observations, we have made the changes in the figure legend, changes have been highlighted.

  1. Figure 1A, lane 1.2.3.4.

" lane " not marked

Thank you for your observation, the appropriated changes were made in the figure.

  1. Supplementary materials not provided

It was provided as an independent file.

  1. Caption to Fig3 "The accession numbers of 55 proteins". The names of proteins are given, not accession numbers. Very poor quality of the figure

The figure is correct showing protein names. Accordingly, the figure legend reads as follows “The accession number of 55 proteins, …………, were input into the STRING 12.0 platform.

Thank you for the observation. The quality of the figure has been improved.

  1. The conclusion "Our findings provide compelling evidence that M. bovis BCG initiates a signaling cascade culminating in the PKC-mediated activation of mTORC1, thereby imposing negative regulation on autophagy" is not well supported. The authors demonstrate phosphorylation of certain proteins, followed by a good literature review of the functions of these proteins, and conclude compelling evidence based on this. The authors may suggest such a relationship, but do not claim that the evidence is compelling

We totally agree with you point of view, thus we have tune down our conclusion in the text.

Reviewer 2 Report

Comments and Suggestions for Authors

This study profiles the phospho-proteomic changes associated with Mycobacterium bovis BCG infection of THP-1-differentiated cells, and also compares the effect of administering a generic protein kinase c (PKC) inhibitor, Gö-6983, on the phosphorylation changes associated with BCG infection. In comparing the phospho-proteomic profiles of the infection with vs. without the PKC inhibitor, the authors highlight proteins associated with phagocytosis and autophagy as important for infection.

The authors have generated a rich phospho-proteomic dataset profiling BCG infection. The manuscript would benefit from additional details clarifying the methodology.

Major Concerns:

-          It is not clear from what was written in the manuscript what the specificity and activity of the PKC inhibitor (Gö-6983) was. Are there any kinases outside of the PKC family that are inhibited by this compound? If so, how many of those kinases are there, and to what extent are their activities inhibited? Additionally, within the PKC family, how many of the kinases are inhibited and to what extent are their activities impacted? It would be helpful to report what is already known in the literature here to provide appropriate context, and this information would be useful in informing the interpretation of whether the phosphorylation changes are necessarily dependent on PKC activity or whether other kinases could be involved in the differential behavior.

-          Based on the description of the infection setup, it seems that THP-1 cells were exposed to BCG for only 30 mins prior to sample processing. Given the infection conditions, how many bacteria successfully infected the THP-1 cells at this timepoint? Do most of the THP-1 cells have a bacterium within them, or are most of the cells bystanders? It would be helpful to see a flow cytometry profile or at least plating for colony forming units under identical infection conditions to get a sense for how many THP-1 cells are infected, since this would influence the interpretation of the resulting proteomic and phospho-proteomic changes (i.e. whether the changes are a result of direct macrophage infection, or whether the changes result largely from bystander signaling from the minority of neighboring infected THP-1 cells or detection of soluble molecules from the bacteria outside of the THP-1 cells).

-          It was not clear how the STRING analysis was performed and interpreted based on what was described in the methods section or Figure 3? How were the two distinct groups identified? Figure 3 appears to show that approximately half of the proteins shown in the figure are interconnected into a shared network, and the other half have either no direct interactions with the rest of the proteins in the visualization, or are interacting with only one other protein in the visualization. Under these circumstances, it is not clear what the groupings are. It is also not clear why only the 55 proteins with Gene Ontology annotations were included in this analysis, rather than the set of 153 proteins with differential phosphorylation, and why interactions between these proteins were selected to be a metric for prioritization for subsequent interpretation.

Minor Concerns:

-          Typo on line 75, should be IFN-g not INF-g.

-          There appears to be a typo in the Figure 1 caption, which makes a reference to “TPA”, which I think should be PMA, since there is no other mention of “TPA” in the manuscript.

Author Response

Reviewer 2

This study profiles the phospho-proteomic changes associated with Mycobacterium bovis BCG infection of THP-1-differentiated cells, and also compares the effect of administering a generic protein kinase c (PKC) inhibitor, Gö-6983, on the phosphorylation changes associated with BCG infection. In comparing the phospho-proteomic profiles of the infection with vs. without the PKC inhibitor, the authors highlight proteins associated with phagocytosis and autophagy as important for infection.

The authors have generated a rich phospho-proteomic dataset profiling BCG infection. The manuscript would benefit from additional details clarifying the methodology.

Major Concerns:

-          It is not clear from what was written in the manuscript what the specificity and activity of the PKC inhibitor (Gö-6983) was. Are there any kinases outside of the PKC family that are inhibited by this compound? If so, how many of those kinases are there, and to what extent are their activities inhibited? Additionally, within the PKC family, how many of the kinases are inhibited and to what extent are their activities impacted? It would be helpful to report what is already known in the literature here to provide appropriate context, and this information would be useful in informing the interpretation of whether the phosphorylation changes are necessarily dependent on PKC activity or whether other kinases could be involved in the differential behavior.

Thank you very much for your concerns regarding the specificity of the PKC inhibitor. Indeed, Go6983 is a reversible and ATP-competitive inhibitor of protein kinase C (PKC) that inhibits several PKC isoformes (IC50 = 7 nM for PKCα and PKCβ; 6 nM for PKCγ; 10 nM for PKCδ; and 60 nM for PKCζ). Gö 6983 does not effectively inhibit PKCµ (IC50 = 20 µM) and can thus be used to differentiate PKCµ from other PKC isozyme. Thus, it is possible that not all identified proteins are specific PKCα and PKCβ substrates. However, Go 6983 has not been reported to exert any effective inhibition on other kinases at the concentration we used in the assay. We have acknowledge this important point in the manuscript text.

-          Based on the description of the infection setup, it seems that THP-1 cells were exposed to BCG for only 30 mins prior to sample processing. Given the infection conditions, how many bacteria successfully infected the THP-1 cells at this timepoint? Do most of the THP-1 cells have a bacterium within them, or are most of the cells bystanders? It would be helpful to see a flow cytometry profile or at least plating for colony forming units under identical infection conditions to get a sense for how many THP-1 cells are infected, since this would influence the interpretation of the resulting proteomic and phospho-proteomic changes (i.e. whether the changes are a result of direct macrophage infection, or whether the changes result largely from bystander signaling from the minority of neighboring infected THP-1 cells or detection of soluble molecules from the bacteria outside of the THP-1 cells).

Our objective was to identify proteins phosphorylated by PKC upon BCG interaction with macrophage cell surface receptors during the early stages of the infectious process. Consequently, we did not assess the infection rate under our experimental conditions. However, at the designated analysis time point (30 minutes), it is anticipated that most bacteria have already engaged cell surface receptors and are likely in the initial stages of the phagocytic process. This assumption is supported by other experiments conducted in the laboratory aimed at determining mycobacterium survival. In these experiments, macrophages are exposed to bacteria for 1 hour, followed by the removal of non-internalized bacteria, revealing that over 90% of macrophages contain internalized bacterium. Given this context, we hypothesize that the observed changes in protein phosphorylation levels primarily result from BCG interaction with macrophage cell surface receptors. However, we acknowledge the possibility that soluble factors secreted by BCG may trigger PKC activation. Nevertheless, if this were the case, it would suggest that BCG primes the macrophage for infection even before direct contact occurs.

      It was not clear how the STRING analysis was performed and interpreted based on what was described in the methods section or Figure 3? How were the two distinct groups identified? Figure 3 appears to show that approximately half of the proteins shown in the figure are interconnected into a shared network, and the other half have either no direct interactions with the rest of the proteins in the visualization, or are interacting with only one other protein in the visualization. Under these circumstances, it is not clear what the groupings are. It is also not clear why only the 55 proteins with Gene Ontology annotations were included in this analysis, rather than the set of 153 proteins with differential phosphorylation, and why interactions between these proteins were selected to be a metric for prioritization for subsequent interpretation.

The rationale for selecting 55 out of the 153 PKC phosphosubstrates that exhibited changes upon BCG infection is elaborated in lines 200 to 207 of the text. We specifically focused on these 55 proteins because GO analysis indicated their involvement in processes previously demonstrated to be regulated by mycobacteria to facilitate survival and infection. The STRING results presented are grounded in data revealing functional, physical, and genetic interactions among different proteins. Therefore, proteins lacking demonstrated interactions are devoid of available data supporting such interactions. However, the absence of documented interactions does not imply an absence of contribution to the assigned gene ontology functions. Conversely, proteins exhibiting clear interactions provide additional support for our findings, strongly suggesting their involvement in macrophage infection regulated by PKC phosphorylation.

Minor Concerns:

-          Typo on line 75, should be IFN-g not INF-g.

Thank you for observation. We have corrected the text.

-          There appears to be a typo in the Figure 1 caption, which makes a reference to “TPA”, which I think should be PMA, since there is no other mention of “TPA” in the manuscript.

Thank you for your observation. We have corrected this mistake.

Reviewer 3 Report

Comments and Suggestions for Authors

Authors tried to identify additional PKC targets that may help M. bovis to survive inside the macrophages. Authors extracted proteins from unstimulated and M. bovis infected THP-1 cells in the presence or absence of a general PKC inhibitor. Mass spectrometry (LC-MS/MS) analysis revealed 736 phosphorylated proteins, among which 153 exhibited alterations in their phosphorylation profiles in response to infection in a PKC-dependent manner. Out of 153 proteins, 55 are involved in various cellular processes, including endocytosis, vesicular traffic, autophagy, and programmed cell death. Authors suggest that PKC may negatively regulate autophagy by phosphorylating proteins within the mTORC1 28 pathway that may promote M. bovis BCG infection of macrophage. The findings are interesting, and authors should address the following concern.

1.       Authors did not present any invitro data to demonstrate the M. bovis BCG survival inside the macrophages in presence of or absence of PKC inhibitor. This experiment is primary and very important to understand the physiological role of PKC activation and their relation to cellular process such as autophagy, and programmed cell death.

2.       Authors carried out experiment with M. bovis BCG to study the PKC-mediated signaling pathways and their role in controlling key cellular process. However, authors generalized their finding with M. tuberculosis infection throughout the manuscript including conclusion sections. It is speculative. M. tuberculosis antigens such as ESAT-6 and CFP10 play an important role in the immune evasion and blocking of autophagy of infected cells, which absent in M. bovis BCG. This should be discussed adequately.

3.       Authors mentioned that THP-1 cells were differentiated to macrophages using PMA. However, these cells are stimulated again with PMA and ionomycin and used as uninfected control.  What is the role of second stimulation. It is not clear.

4.       Authors used only one MOI (1:1), which is not enough to get infection of all cells. Did you check phagocytic index of at different MOI and time points to find optimum condition for proper infection?

5.       Section 2.5 to 2.6.6: These sections were provided as review article. The subtitle should be changed to highlight your major findings. These sections contain a lot of speculative data and should be reduced significantly.

6.       The results and discussion sections were written together. Does it meet journal policy? There is no section 3.

7.       Line #75: not INF, it is IFN.

8.       Figure 3 is not clear and does not meet publication quality.

9.       Line #581-582: ‘Differentiated THP-1 581 macrophages (2.5 × 10^5) were activated with M. bovis BCG’. Not activated, macrophages were infected with M. bovis BCG.

Author Response

Reviewer 3

Authors tried to identify additional PKC targets that may help M. bovis to survive inside the macrophages. Authors extracted proteins from unstimulated and M. bovis infected THP-1 cells in the presence or absence of a general PKC inhibitor. Mass spectrometry (LC-MS/MS) analysis revealed 736 phosphorylated proteins, among which 153 exhibited alterations in their phosphorylation profiles in response to infection in a PKC-dependent manner. Out of 153 proteins, 55 are involved in various cellular processes, including endocytosis, vesicular traffic, autophagy, and programmed cell death. Authors suggest that PKC may negatively regulate autophagy by phosphorylating proteins within the mTORC1 28 pathway that may promote M. bovis BCG infection of macrophage. The findings are interesting, and authors should address the following concern.

  1. Authors did not present any invitro data to demonstrate the M. bovis BCG survival inside the macrophages in presence of or absence of PKC inhibitor. This experiment is primary and very important to understand the physiological role of PKC activation and their relation to cellular process such as autophagy, and programmed cell death.

Thank you for your pertinent comment, indeed those experiments are key to link PKC activity resulting from BCG infection autophagy and bacterium survival.  However, the data presented here is intended to inspire hypotheses and drive research into the understanding of how mycobacteria exploit the macrophage machinery to survive and seed a successful infection. Nonetheless, there is a strong background showing that PKC is implicated in various macrophage functions such as phagocytosis, maturation of phagosome, apoptosis and the production of cytokines/chemokines/immune effector molecules (Zheleznyak & Brown; 1992, Holm et al; 2003 and Yan Hing et al; 2004). Additionally, it has been recently shown that inhibition of PKC (specifically the alpha isoform) has been shown to promote autophagy activation (Wong, V.K.W. et al 2017).  On the other hand, experimental evidence clearly supports that mycobacterium infection impair autophagy (Kumar et al; 2010, Shui et al; 2011, Seto et al; 2012 and Vergne et al; 2014). Thus, our data points out to PKC as the missing link between these two processes during macrophage infection by mycobacteria.

Supporting 2.       Authors carried out experiment with M. bovis BCG to study the PKC-mediated signaling pathways and their role in controlling key cellular process. However, authors generalized their finding with M. tuberculosis infection throughout the manuscript including conclusion sections. It is speculative. M. tuberculosis antigens such as ESAT-6 and CFP10 play an important role in the immune evasion and blocking of autophagy of infected cells, which absent in M. bovis BCG. This should be discussed adequately.

Thank you for your very important comment.

We have acknowledged that our data generated using M. bovis BCG cannot be extrapolated to M. tuberculosis thus we have modified the concluding remarks and conclusion sections. Changes are highlighted.

  1. Authors mentioned that THP-1 cells were differentiated to macrophages using PMA. However, these cells are stimulated again with PMA and ionomycin and used as uninfected control.  What is the role of second stimulation. It is not clear.

PMA, a phorbol ester, mimics the function of diacylglycerol, activating the PKC pathway. In-vitro differentiation of THP-1 monocytes to macrophages is induced by activating the PKC pathway with PMA for 24 hr (Tsuchiya et al., 1982). However, it has been reported that some PKC isoforms are downregulated in response to prolonged exposure to PMA (Chen et al., 1997; Isonishi et al., 2000). In line with this, experiments conducted in our laboratory, although unpublished, indicate that PKC isoforms recover after 96 hrs following THP-1 monocyte stimulation for differentiation. Consequently, we differentiated macrophages for 72 hr following a 12-hour deprivation of growth factors to prevent proteins' phosphorylation induced by growth factors present in serum. Thus, macrophages were prepared to respond to PMA and Ionomycin re-stimulation and the PKC signaling pathway. These stimuli were consistently employed as a positive control in all our experiments.

  1. Authors used only one MOI (1:1), which is not enough to get infection of all cells. Did you check phagocytic index of at different MOI and time points to find optimum condition for proper infection?

Our objective was to identify proteins phosphorylated by PKC upon BCG interaction with macrophage cell surface receptors during the early stages of the infectious process. Consequently, we did not assess the infection rate under our experimental conditions. However, at the designated analysis time point (30 minutes), it is anticipated that most bacteria have already engaged cell surface receptors and are likely in the initial stages of the phagocytic process. This assumption is supported by other experiments conducted in the laboratory aimed at determining mycobacterium survival. In these experiments, macrophages are exposed to bacteria for 1 hour, followed by the removal of non-internalized bacteria, revealing that over 90% of macrophages contain internalized bacterium. Given this context, we hypothesize that the observed changes in protein phosphorylation levels primarily result from BCG interaction with macrophage cell surface receptors.

  1. Section 2.5 to 2.6.6: These sections were provided as review article. The subtitle should be changed to highlight your major findings. These sections contain a lot of speculative data and should be reduced significantly.

Thank you for your comment.  The data discussed in this section is intended to inspirate hypotheses drive research into the understanding how mycobacteria coopt the macrophage machinery to survive and seed a successful infection. The length of these section results from the in deep analysis combining our findig with those of the literature and although speculative we consider they will be interesting to researches in the field to persue new venues in the understanding of the complex molecular relationship established by mycobacterium and one of its main host cells, the macrophage.

  1. The results and discussion sections were written together. Does it meet journal policy? There is no section 3.

To avoid been repetitive and for the sake of length we combined the results and discussion.

  1. Line #75: not INF, it is IFN.

Thank you for your comment. We have corrected this mistake.

  1. Figure 3 is not clear and does not meet publication quality.

 We have now provided a high-definition picture.

  1. Line #581-582: ‘Differentiated THP-1 581 macrophages (2.5 × 10^5) were activated with M. bovis BCG’. Not activated, macrophages were infected with M. bovis BCG.

Thank for the observation, we have changed the text.